

# GPR18 undergoes a high degree of constitutive trafficking but is unresponsive to N-Arachidonoyl Glycine

David B. Finlay[1], Wayne R. Joseph[2], Natasha L. Grimsey[1,*] and Michelle Glass[1,*]

[1] Centre for Brain Research and Department of Pharmacology and Clinical Pharmacology, Faculty of Medical and Health Sciences, University of Auckland, Auckland, New Zealand
[2] Auckland Cancer Society Research Centre, Faculty of Medical and Health Sciences, University of Auckland, Auckland, New Zealand
[*] These authors contributed equally to this work.

## ABSTRACT

The orphan receptor GPR18 has become a research target following the discovery of a putative endogenous agonist, N-arachidonoyl glycine (NAGly). Chemical similarity between NAGly and the endocannabinoid anandamide suggested the hypothesis that GPR18 is a third cannabinoid receptor. GPR18-mediated cellular signalling through inhibition of cyclic adenosine monophosphate (cAMP) and phosphorylation of extracellular signal-regulated kinase (ERK), in addition to physiological consequences such as regulation of cellular migration and proliferation/apoptosis have been described in response to both NAGly and anandamide. However, discordant findings have also been reported. Here we sought to describe the functional consequences of GPR18 activation in heterologously-expressing HEK cells. GPR18 expression was predominantly intracellular in stably transfected cell lines, but moderate cell surface expression could be achieved in transiently transfected cells which also had higher overall expression. Assays were employed to characterise the ability of NAGly or anandamide to inhibit cAMP or induce ERK phosphorylation through GPR18, or induce receptor trafficking. Positive control experiments, which utilised cells expressing hCB1 receptors (hCB1R), were performed to validate assay design and performance. While these functional pathways in GPR18-expressing cells were not modified on treatment with a panel of putative GPR18 ligands, a constitutive phenotype was discovered for this receptor. Our data reveal that GPR18 undergoes rapid constitutive receptor membrane trafficking—several-fold faster than hCB1R, a highly constitutively active receptor. To enhance the likelihood of detecting agonist-mediated receptor signalling responses, we increased GPR18 protein expression (by tagging with a preprolactin signal sequence) and generated a putative constitutively inactive receptor by mutating the hGPR18 gene at amino acid site 108 (alanine to asparagine). This A108N mutant did cause an increase in surface receptor expression (which may argue for reduced constitutive activity), but no ligand-mediated effects were detected. Two glioblastoma multiforme cell lines (which endogenously express GPR18) were assayed for NAGly-induced pERK phosphorylation, with negative results. Despite a lack of ligand-mediated responses in all assays, the constitutive trafficking of GPR18 remains an interesting facet of receptor function and will have consequences for understanding the role of GPR18 in physiology.

Corresponding author
Michelle Glass,
m.glass@auckland.ac.nz

## INTRODUCTION

The orphan G protein-coupled receptor (GPCR) GPR18 has been postulated to be a member of the cannabinoid receptor family, and is a subject of research interest for therapeutic applications as diverse as regulation of intraocular pressure, cancer, and immune system regulation (*Caldwell et al., 2013*; *Qin et al., 2011*; *Takenouchi et al., 2012*). GPR18 mRNA is present most abundantly in the testis, spleen, lymph nodes and peripheral blood leukocytes (*Gantz et al., 1997*; *Samuelson, Swanberg & Gantz, 1996*). *Kohno et al. (2006)* found GPR18 mRNA to be highly expressed in several human T-Cell lymphotrophic virus-transformed cell lines and primary human peripheral lymphocyte subsets, with very high expression identified in phytohaemagglutinin-activated CD4+ T-cells. A putative endogenous ligand for GPR18 was discovered by screening a bioactive lipid library against polyclonal L929 cells (a mouse connective tissue cell line) either stably expressing human GPR18 (hGPR18) or mock transfected, and measuring intracellular calcium flux. The lipid N-arachidonoyl glycine (NAGly; 10 $\mu$M) was identified as a hit in the screen and also induced significant $Ca^{2+}$ fluxes in Chinese hamster ovary (CHO-K1) and K562 (myelogenous leukaemia) cells stably transfected with GPR18. Further studies in hGPR18-transfected CHO cells demonstrated a highly potent, pertussis toxin-sensitive inhibition of forskolin-induced cyclic adenosine monophosphate (cAMP) levels ($IC_{50}$ 20 nM) (*Kohno et al., 2006*).

Despite the structural similarity of NAGly to the endogenous cannabinoids anandamide and 2-arachidonyl glycerol, NAGly shows no measurable affinity for either the type one (CB1R) or type two (CB2R) cannabinoid receptors (*Bradshaw, Lee & McHugh, 2009*; *Sheskin et al., 1997*). Conversely, other cannabinoids (some with affinity for CB1R and CB2R) have been proposed as GPR18 ligands. Boyden chamber cell migration assays indicated the phytocannabinoid $\Delta^9$-tetrahydrocannabinol ($\Delta^9$-THC) and the synthetic cannabinoids abnormal cannabidiol (AbnCBD) and O-1602 to be GPR18 agonists along with NAGly (*McHugh et al., 2010*; *McHugh et al., 2012a*). In turn, this migratory effect could be attenuated by treatment with the AbnCBD receptor antagonist O-1918 and partial agonist/antagonist cannabidiol in both BV-2 cells endogenously expressing mouse GPR18 (mGPR18) and in hGPR18-transfected human embryonic kidney (HEK) cells (*McHugh et al., 2010*; *McHugh et al., 2012a*). Also in HEK-hGPR18 cells, NAGly ($EC_{50}$ 44.5 nM), O-1602 (65.3 nM), AbnCBD (836nM), $\Delta^9$-THC (960 nM), anandamide (AEA; 3.83 $\mu$M) and arachidonoylcyclopropylamide (ACPA; 51.1 $\mu$M) all acted as full agonists at GPR18 when screened for the ability to activate Extracellular Signal-Regulated Kinase (pERK), while cannabidiol and AM251 (a CB1R antagonist) acted as partial agonists/antagonists (*McHugh et al., 2012a*). To date, one report has also described a constitutive (ligand independent) phenotype for GPR18 in a yeast model of receptor function; alanine to asparagine mutation of a residue in the GPR18 sequence (at amino acid position 108) attenuated GPR18 signalling-mediated proliferation of histidine auxotrophic yeast, in the absence of agonists (*Qin et al., 2011*).

One of the most significant factors impeding the deorphanisation of GPR18 is an apparent disconnect between the tissue localisations of GPR18 and its putative selective

ligand, NAGly. hGPR18 mRNA has been reported to have moderate to high expression in the testes, peripheral blood leukocytes, thyroid, lungs and specific brain regions (hypothalamus, cerebellum, brain stem and striatum), and low to moderate expression in the spleen, thymus, ovaries, uterus, stomach and intestines. Many other tissues including the brain (amygdala, frontal cortex, hippocampus, cerebellum, thalamus and brain stem), heart, lungs, liver, kidney, pancreas, colon, skeletal muscle, skin, placenta, prostate, adrenal medulla and adrenal cortex exhibited low or undetectable hGPR18 mRNA (*Gantz et al., 1997*; *Vassilatis et al., 2003*). In contrast, in rats the highest levels of NAGly were measured in the spinal cord ($\sim$140 pmol g$^{-1}$ dry weight), small intestine ($\sim$130 pmol g$^{-1}$ dry weight), kidneys (100 pmol g$^{-1}$ dry weight) and skin ($\sim$50 pmol g$^{-1}$ dry weight) (*Huang et al., 2001*). NAGly is much less abundant in organs with moderate to high GPR18 expression; the testes and spleen in particular contained low levels (approximately 30 and 5 pmol g$^{-1}$ dry weight, respectively) (*Huang et al., 2001*). It should be noted, however, that such co-localisation data are of ambiguous significance; the endocannabinoids AEA and 2-arachidonyl glycerol are known to be rapidly synthesised on demand (*Pacher & Kunos, 2013*) and the same may be true for NAGly, implying that it might only be present when an *in vivo* system actively induces its synthesis.

In addition to the apparent mismatch in GPR18 and NAGly localisation *in vivo*, two other recent publications present evidence in opposition to the candidacy of GPR18 as a NAGly receptor. *Yin et al. (2009)*, used $\beta$-arrestin recruitment as a measure of receptor activity utilising ligands from bioactive lipid, endocannabinoid and orphan ligand libraries, as well as NAGly, and found that none of the ligands tested activated the receptor. Subsequently, *Lu, Puhl & Ikeda (2013)* undertook a thorough characterisation of mGPR18-mediated signalling in rat sympathetic neurons; NAGly, AEA and AbnCBD were not found to inhibit Ca$^{2+}$ currents, modulate cAMP levels, or activate G protein-coupled inwardly rectifying potassium channels. Five different GPR18 receptor mutants were created, in the hope that they might alter the constitutive activity of the receptor in a manner detectable as changes in Ca$^{2+}$ currents. These also failed to result in detectable signalling changes. Finally, in case the negative assay results were the consequence of different G protein-coupling, G$\alpha$s, G$\alpha$z and G$\alpha_{15}$ G proteins were each transfected into the recipient cells of interest, but these also failed to induce detectable Ca$^{2+}$ current changes when the cells were stimulated with the reported GPR18 ligands (*Lu, Puhl & Ikeda, 2013*).

Our study set out to investigate hGPR18 trafficking under constitutive and ligand-stimulated conditions, but given the conflicting evidence around ligand recognition, we initially aimed to confirm the ability of putative GPR18 ligands to activate previously described signalling pathways (cAMP and pERK). Here we report that hGPR18 exhibits a highly constitutively active trafficking phenotype, but fails to respond to previously-reported putative ligands in either transfected cells, or cells endogenously expressing the receptor.

## MATERIALS AND METHODS

### Drugs and chemicals

AEA, O-1602 and AbnCBD were purchased from Cayman Chemical Company (Ann Arbor, MI, USA). NAGly was bought from both Cayman and Enzo Biosciences (in order to exclude the possibility that lack of responses were related to a specific company's synthesis/production; results were equivalent for both suppliers). Forskolin, CP55, 940, SR141716A, AM251 and AM630 were purchased from Tocris Bioscience (Bristol, UK). Phorbol-12-myristate-13 acetate (PMA) was purchased from Sigma Aldrich (St Louis, MO, USA). U0126 was purchased from Cell Signaling Technology (Danvers, MA, USA). All cannabinoid and lipid ligands were diluted in ethanol prior to storage in single use aliquots at $-80$ °C. The lipophilic properties of cannabinoids and lipid ligands such as NAGly cause them to adsorb onto the hydrophobic surfaces of plasticware and interact with serum components. To sustain drug availability in assays, the vessels in which cannabinoids were diluted were silanised (Coatasil, ThermoFisher AJA2293; Waltham MA, USA) and autoclaved prior to use, and assay media or HBSS were supplemented with 1 mg/ml bovine serum albumin (BSA, MP Biomedicals ABRE, Santa Ana, CA, USA).

### Cloning, plasmid preparation and sequence verification

A plasmid incorporating the gene for amino-terminally haemagglutinin (HA) tagged hGPR18 (HA-hGPR18) in a pCAG cloning vector variant was kindly gifted by Associate Professor Heather Bradshaw (University of Indiana, Bloomington, IN, USA) and the sequence for the inserted gene was verified (Massey Genome Service, Palmerston North, NZ). In order to facilitate establishing a "Flp-in" cell line stably expressing HA-hGPR18, the gene was re-ligated into pcDNA™ 5/FRT (Life Technologies Corporation, Carlsbad, CA, USA) utilising standard DNA cloning methods, and was sequence verified.

In order to generate the putative constitutively inactive form of hGPR18 (A108N, 108 refers to amino acid residue 108 in the hGPR18 sequence *Qin et al., 2011*); a three base mutation (GCT to AAC) in the HA-hGPR18 gene (in pcDNA™ 5/FRT) was generated using the QuikChange® (Stratagene) strategy for site-directed mutagenesis but utilising KAPA HiFi HotStart DNA polymerase (KAPA Biosystems, Wilmington, MA, USA).

With the aim of enhancing GPR18 expression and detection, a 3HA amino terminal-tagged version of hGPR18 in pcDNA™ 5/FRT was created, incorporating a 5′ bovine pre-prolactin signal sequence (pplss). This 30 amino-acid sequence is recognised by the signal recognition particle and thereby increases nascent protein translocation across the endoplasmic reticulum (ER) membrane for considerably increased secretion efficiency (*Kurzchalia et al., 1986*), but (like typical signal peptides) is cleaved prior to surface expression (*Belin et al., 1996*). The pplss-3HA-hGPR18 construct was generated using the original HA-hGPR18 construct, with the pplss-3HA fragment added at the 5′ end of the coding region (replacing the original HA sequence) using standard DNA cloning methods. The purpose of increasing the number of HA-tags from one to three was to increase the

intensity of staining for improved sensitivity of assays utilising immunocytochemistry readouts.

All work employing genetically modified organisms was performed in compliance with the University of Auckland Biological Safety Committee (under the New Zealand Environmental Protection Authority) approval number B00343-GMO08-UA003 (s67a).

## Cell culture

HEK-FT cells were received from Associate Professor K Pfleger (University of Western Australia), HEC-1a cells were received from Professor L Chamley (University of Auckland), CHO-K1 cells were from American Type Culture Collection (ATCC CRL-9618, Manassas VA), and human primary glioblastoma multiforme (GBM) cells (NZB11 and NZB19) were received from the Auckland Cancer Society Research Centre (used with the permission of Distinguished Professor B Baguley, University of Auckland, and in compliance with ethical approval ALK/2000/AM03). HEK Flp-In$^{TM}$ -293 cells were from Life Technologies Corporation (R750-07). Cells were routinely cultured in polystyrene vented-cap flasks in a humidified atmosphere of 5% $CO_2$ in air; except the GBM lines which were cultured in 5% $CO_2$ under normoxic conditions (5% oxygen; $O_2$ displaced with $N_2$ gas). All cell culture reagents were from Life Technologies unless noted. All HEK cell lines were cultured in Dulbecco's Modified Eagle's Medium (DMEM, Life Technologies 11995-073) supplemented with 10% v/v foetal bovine serum (FBS, Sigma Aldrich 12203C; St. Louis, MO, USA), CHO-K1 and HEC-1a cells were cultured in DMEM/F12 medium (Life Technologies 11330-057) supplemented with 10% v/v FBS, and the GBM cell lines were cultured in $\alpha$MEM (Life Technologies 12561-056) supplemented with 10% v/v FBS. Stably-transfected cells were maintained with selection antibiotics: HEK Flp-in wild-type (WT) 100 μg/ml Zeocin$^{TM}$ (Life Technologies R250-01); HA-hGPR18 and HA-TCS-hCB1 ('TCS' refers to a thrombin cleavage site between the HA tag and receptor N-terminus which was inconsequential to these experiments) in HEK Flp-in 100 μg/ml Hygromycin B (Life Technologies 10687-010); HEK-FT 400 μg/ml G418 (Life Technologies 11811-031); 3HA-hCB1 HEK 250 μg/mL Zeocin$^{TM}$ (*Cawston et al., 2013*).

## Generation of stable cell lines and transfection

Transfections for stable expression in HEK Flp-in cells were performed using the Flp-in System$^{TM}$ into HEK Flp-in WT cells, according to an optimised version of the manufacturer's instructions. In brief, low passage HEK Flp-in WT cells were seeded in a 24 well plate at a density of 350,000 cells per well and were cultured in normal growth medium (DMEM + 10% FBS) for 24–36 h until 80–90% confluent. A transfection mixture was prepared, comprising 0.4 μg per well each of the plasmid-of-interest (in pcDNA$^{TM}$ 5/FRT) and pOG44, and Lipofectamine 2000, in Opti-MEM®. Transfection and selection were performed according to the manufacturer's instructions, using a selection concentration of Hygromycin B of 100 μg/ml. Cells were perpetually cultured in Hygromycin B to ensure persistent transgene expression.

Initial characterisation by immunocytochemistry indicated that HEK Flp-in cells stably expressing HA-hGPR18 expressed the receptor non-homogeneously and at low levels.

Therefore, the population was sorted by Fluorescence Activated Cell Sorting (FACS) in an attempt to enrich for a population of cells expressing receptor more uniformly. Cells in culture were detached from the flask with Versene, resuspended and triturated thoroughly in normal growth medium, and transferred to a 15 ml tube. Cells were counted and treated with primary mouse anti-HA antibody (diluted 1:500; Covance MMS-101P; Princeton, NJ, USA) for 30 min at 37 °C, agitating the tube regularly. Cells were washed twice in normal growth medium, and then labelled with secondary goat anti-mouse-Alexa-488 antibody (diluted 1:300; Life Technologies A11029). After washing, cells were resuspended in serum-free phenol red-free DMEM and 200 U/ml penicillin/streptomycin at a final cell concentration of $\sim 1.5e^6$ cells/ml. To ensure that cells were in single-cell suspension, this mixture was passed through a 70 μm cell strainer and immediately taken to the flow cytometer (FACSVantage Cell Sorter; Becton Dickinson, Franklin Lakes, NJ, USA) for sorting. The cells with the highest 10% of fluorescent labelling were sorted into a collection vessel containing normal growth medium (DMEM + 10% FBS) and 200 U/ml penicillin/streptomycin, and were thereafter propagated for subsequent use in experiments.

## Assay for cAMP with CAMYEL

Generation of cAMP was detected using a Bioluminescence Resonance Energy Transfer (BRET) assay as previously described (*Cawston et al., 2013*), but using an in-well transfection method. Briefly, on day one, cells were seeded in a Poly-D-Lysine (PDL, Sigma Aldrich)-treated 96-well white plate (CulturPlate-96, PerkinElmer 6005688; PerkinElmer, Watham, MA, USA) in addition to a parallel PDL-treated 96-well transparent cell culture plate. Cells were seeded to be 60–80% confluent following 18–24 h in culture. Cells were then transiently transfected. HEK-FT and CHO-K1 cells were co-transfected with two plasmids; one encoding receptor and the other encoding the CAMYEL biosensor (ATCC MBA-277). HEK cells stably expressing receptor were transiently transfected with the plasmid encoding CAMYEL only. When co-transfection was performed, the total amount of DNA per well was not altered (0.8 μg) but DNA ratios were changed so that receptor DNA was twice as abundant as biosensor (to increase likelihood of reporter/CAMYEL-expressing cells expressing the protein of interest). Approximately six hours after transfection, medium was changed and cells were cultured for a further two days, to ensure maximum expression of the transiently expressed proteins. On day four, cells were washed with 70 μl per well Hank's Balanced Salt Solution (HBSS, Life Technologies 14025-134) pre-warmed to 37 °C, and then serum-starved for 30 min in 70 μl per well HBSS+1mg/ml BSA (37 °C). Cells were then treated with 5 μM Coelenterazine-H for 6 min prior to addition of drugs/vehicle (in HBSS+1 mg/ml BSA). Immediately following drug addition, emission signals were detected with λ 460/30 nm and λ 535/30 nm bandpass filters using a VICTOR$^{TM}$ X Light Luminescence Plate Reader at 37 °C, as previously described (*Cawston et al., 2013*). Up to 20 wells per set were read repeatedly over an elapsed time of 20–25 min.

Data are presented as inverse BRET ratios (λ 460 nm emissions/λ 535 nm emissions) such that an increase in cAMP corresponds to an increased ratio. Data from across the time course were analysed by Area-under-the-curve in GraphPad Prism 6 (GraphPad Software Inc., La Jolla, CA, USA). This analysis used the trapezoid rule to compute total cAMP

responses for each experimental condition over the time course of that specific CAMYEL run. Data were normalised to a matched forskolin (FSK) condition (100%) and vehicle condition (0%), enabling combination of data from independent experiments.

## Quantitative assays for pERK stimulation

Activation of ERK (pERK) was detected quantitatively using an immunocytochemistry method. Briefly, cells were seeded in PDL-treated 96-well cell culture plates (Nunc, ThermoFisher Scientific NUN167008, Waltham, MA, USA). HEK and GBM cells were seeded as described above. For assays on transiently-expressing HEK Flp-in WT cells, transfections were performed 18 h after seeding. Medium was changed 6 h after transfection. Approximately 24 h after seeding (or, for the transiently transfected HEK cells, 24 h after medium change), cells were serum starved for at least 18 h in 60 µl per well serum-free medium (DMEM + 1mg/ml BSA for HEKs, or $\alpha$MEM + 5 mg/ml BSA for GBMs). Drug/vehicle dilutions were made at 2×the desired concentration in serum-free medium and applied in a staggered fashion with the 96-well plates positioned on the surface of a waterbath at 37 °C. At the conclusion of the time course, plates were immediately moved to an ice bed where contents of all wells were aspirated and 4% paraformaldehyde (PFA) was dispensed rapidly to each well. Plate contents were fixed for 10 min, washed twice with phosphate buffered saline (PBS), and then treated with ice-cold 90% methanol for 10 min to unmask the pERK epitope. Methanol was aspirated and plates air-dried for approximately 15 min before being re-hydrated with PBS containing 0.2% Triton X-100 (PBS+ T). Wells were probed with primary rabbit anti-pERK antibody (Cell Signaling Technology 4370; Danvers, MA, USA; diluted 1:200 in immunobuffer comprised of PBS+ T, 1% normal goat serum and 0.4 mg/ml merthiolate) for 3 h at room temperature, or overnight at 4 °C on a rocker. Secondary goat anti-rabbit (Alexa 594 or Alexa 488, Life Technologies) antibody (diluted 1:400 in immunobuffer) was applied to each well for 3 h at room temperature, or overnight at 4 °C, in the dark. Cells that had been transiently transfected during this protocol were also probed for receptor (mouse anti-HA antibody) to confirm receptor expression (see below *Image acquisition and analysis* for transient pERK). Nuclei were stained with Hoechst 33258 (Life Technologies H1398) and plates washed. Image acquisition and analysis was performed as described below to determine fluorescence intensity per cell, where brighter staining was indicative of greater pERK. For each cell type assayed, time course data were normalised to pERK levels induced by the presence of 10 µM of U0126 (0%) and 100 nM PMA (100%) and area-under-the-curve analysis (summary data) was normalised for fold-over basal pERK. Data were plotted using GraphPad Prism 6.

## Assays for receptor constitutive trafficking and expression

Prior to manipulation cells were equilibrated in serum-free medium (SFM) for 30 min at 37 °C. Unless noted antibody incubations and drug stimulations were performed at 37 °C. When cells required manipulation outside the incubator during an assay (for addition of a drug or antibody) plates were placed on a polystyrene surface to reduce temperature changes by heat conduction. At the conclusion of experiments receptor trafficking was

rapidly arrested by chilling assay plates on ice. Following any further requisite antibody incubation, cells were fixed in 4% PFA (10 min at room temperature) and washed twice with PBS.

In order to observe constitutive turnover of surface receptor expression a 'live antibody feeding' approach was utilised (*Grimsey et al., 2008*). This method entailed incubating cells with mouse anti-HA primary antibody (diluted 1:500 in SFM) for 2 h. After chilling assay plates on ice, detection of only surface-resident receptors was achieved by staining with secondary antibody (goat anti-mouse (Alexa 594/Alexa 488), 1:300 in SFM) for 30 min prior to washing with SFM and fixation. As antibodies do not permeate live cells, only receptors resident at the cell surface were labelled. Following fixation with 4% PFA (10 min) and washing with PBS, 'cycled' receptors (receptors that had been on the surface at some point during the live primary antibody feeding period but internalised during this time) were detected by applying a different dye-conjugated secondary antibody (549, if 488 was used to label surface receptor), in the presence of detergent to permeabilise cell membranes (1:400 in immunobuffer).

For experiments designed to measure basal or ligand-perturbed surface receptor expression, mouse anti-HA primary antibody was applied for 45 min at 4 °C (in order to prevent receptor trafficking) after drug treatments. Unbound anti-HA antibody was then washed off, and secondary goat anti-mouse antibody was applied to the live cells (in SFM, for 30 min at 4 °C) prior to washing with SFM and fixation.

In order to measure total receptor expressed by the cell, irrespective of sub-cellular localisation, mouse anti-HA primary antibody was applied after the cells had been fixed and permeabilised. After unbound primary probe was washed off, secondary goat anti-mouse antibody was applied for 3 h at room temperature, or overnight at 4 °C.

Subsequent to antibody labelling, cells were Hoechst stained as above and then microscope images acquired and analysed as described below. In transient expression assays utilising single HA-tagged GPR18, staining was compared to a single HA-tagged hCB1 construct: HA-TCS-hCB1 in pcDNA5$^{TM}$/FRT (also used to generate a stable Flp-in HEK cell line, as described above).

## Image acquisition and analysis

For assays involving quantitative immunocytochemistry (pERK, trafficking, expression), readout was from widefield fluorescence microscopy images acquired by automated microscopy (either Discovery-1$^{TM}$ or ImageXpress$^{®}$ Micro XLS, both Molecular Devices, Sunnyvale, CA, USA; the latter system was utilised for assays on pplss-3HA-hGPR18 HEK Flp-in cells and GBM cells), and associated analysis platforms MetaMorph$^{®}$/MetaXpress$^{®}$ (v6.2r6 or v5.3.05 respectively; Molecular Devices). Acquisition parameters and analysis methods were similar to those described previously (*Grimsey et al., 2008*). In brief, four images were taken from adjacent sites in the centre of each culture plate well using a 10×objective lens and excitation and emission filters appropriate for the fluorophores of interest, selecting exposure times for maximal detection sensitivity while avoiding image/camera saturation. High resolution images shown in the Results were acquired on the same systems, but were acquired utilising a 40×objective.
For intensity analysis, prior to quantitation two processing steps were performed to automate exclusion of image areas which were not of interest or could negatively influence the analysis (not containing intact cells, or containing artefactual bright debris; these steps represent methodological improvements since our 2008 report). Firstly, Hoechst nuclear stain images were thresholded (applied equally across all conditions under comparison) to identify image areas containing nuclei and a binary mask generated. A 'dilation' algorithm was applied to expand this mask area in order to encapsulate the entire cell area ("nuclei mask"). Secondly, images of the fluorophore intended for intensity quantitation (i.e., Alexa 488 or 594) were inspected for any evidence of aberrant bright debris (for example as may arise from secondary antibody precipitation). If present, images were thresholded (applied equally across all conditions for comparison) to encapsulate this bright staining *without* including any genuine/"real" staining. A binary mask was generated and dilated to expand the mask area in order to enclose the area of "blur" which emanates from brightly stained objects but again does not represent genuine staining of interest ("debris mask"). For all fluorophores/wavelengths, only image areas within the nuclei mask but outside the debris mask were quantitated. Images were also assessed qualitatively; those containing low to medium-intensity artefacts not excluded by automated debris exclusion, or those that were out of focus, were excluded from quantitative analysis. Images were then analysed as in *Grimsey et al. (2008)* for total (integrated) fluorescence intensity per cell by utilising a user-defined threshold (applied equally across all conditions for comparison), measuring the sum of all pixel grey levels within this threshold, subtracting background (threshold multiplied by pixels within the thresholded area) and dividing by the cell count in the image. Cell counts were assessed utilising the MetaMorph®/MetaXpress® dropin "Find Spots."

For pERK experiments with transient receptor expression, analysis of pERK was separated into cells expressing or not expressing receptor. This was achieved by applying a threshold to the receptor staining image in order to delineate areas with receptor-positive cells versus receptor-negative cells; corresponding image masks were applied to the nuclei and pERK images and analysed for intensity per cell as described above. These custom MetaMorph®/MetaXpress® journals are available on request. To quantitate the proportion of cells positive for one or more fluorophores/proteins of interest the MetaMorph®/MetaXpress® module "Cell Scoring" or "Multi-wavelength Cell Scoring" was utilised, in accordance with manufacturer recommendations.

### RNA isolation, cDNA synthesis and Reverse-Transcription PCR

Pellets of 500,000 low passage cells were frozen at $-80\,°C$. Subsequent extraction of mRNA was performed using the Quick-RNA™ miniprep kit (Zymo Research Corporation, Irvine, CA, USA) according to manufacturer instructions, and then quantified using a NanoDrop™ 1000 Spectrophotometer (Thermo Scientific, Waltham, MA, USA). Synthesis of cDNA was performed using the qScript Flex cDNA Synthesis kit (Quanta BioSciences, Gaithersburg, MD, USA) primed with a 50:50 mix of oligo d(T) and random primers, according to manufacturer instructions. Reverse-transcription PCR (RT-PCR) was performed according to traditional methods, using primer-optimised annealing

temperatures. Primers used were: human $\beta$-actin (forward GCTCGTCGTCGACAACG-GCTC, reverse CAAACATGATCTGGGTCATCTTCTC, amplicon of 353bp); human CB1 (forward CAGCAGACCAGGTGAACATT, reverse GGTCCACATCAGGCAAAACG, amplicon of 512bp); human CB2 (forward GACCTTCACAGCCTCTGT, reverse TCC-CATCAGCCTCTGTCT, amplicon of 689bp); human GPR18 (forward CCACCAAGAA-GAGAACCAC, reverse GAAGGGCATAAAGCAGACG, amplicon of 596bp); and human GPR55 (forward AGTTTGCAGTCCACATCC, reverse GCGCTCCAGGTATCATC, amplicon of 469bp). PCR products were run alongside a 1Kb+ ladder (Life Technologies Corporation, Carlsbad CA) on a gel of 1% agarose (TAE), and stained with RedSafe$^{TM}$ Nucleic Acid Stain (iNtRON Biotechnology, Korea). Gel imaging was performed on a ChemiDoc reader (Bio-Rad Laboratories, Hercules CA).

### Data analysis

All datasets were tested for compliance with parametric test assumptions of normality and equality of variance. 1-way repeated measures ANOVA tests were performed on the means from at least 3 independent experiments (*Lew, 2007*) or as noted. Where statistically significant differences were found over the entire group being tested ($p < 0.05$), Holm-Šídák post-hoc testing was performed. If the population failed either the normality or equality of variance tests, data were transformed so as to meet parametric test assumptions. If tests for normality or equality of variance still failed, non-parametric repeated measures ANOVA on ranks tests were performed, with Dunn's post-test if an overall statistically significant difference was found. Post-hoc significance ($p < 0.05$) is graphically illustrated by the * symbol. Brackets indicate the groups found to be significantly different from a matched control (the left-most bar under the bracket). Statistical analysis was performed using SigmaPlot$^{TM}$ 11.0 or 12.5 (Systat Software Inc., San Jose, CA, USA).

## RESULTS

### Generation of cell lines stably expressing hGPR18

Expressing HA-hGPR18 in HEK cells using the Flp-in system was expected to result in the gene of interest being isogenically incorporated into cells, thereby producing a cell line where expressing cells exhibit matched expression. However, immunocytochemistry revealed a wide range of expression levels, with many antibiotic-resistant cells expressing negligible receptor below detection limits (data not shown). No antibody labelling was detected in untransfected cells (data not shown), confirming the specificity of the anti-HA antibody. In order to enrich for a more homogeneously-expressing cell population with detectable cell surface receptor expression, FACS was employed to collect the 10% highest expressing cells. This resulted in a population of cells expressing low levels of surface receptor, but moderate levels of total expression which was uniformly punctate and predominantly intracellularly-localised (see Figs. 1A and 1B). By comparison, total expression of HA-TCS-hCB1 in stably transfected Flp-in HEK cells was overall higher (Fig. 1D), with high levels of surface receptor also (Fig. 1E).

Low ratios of surface to intracellular receptor expression can be an indication of high levels of constitutive trafficking (*Miserey-Lenkei et al., 2002*; *Uwada et al., 2014*).

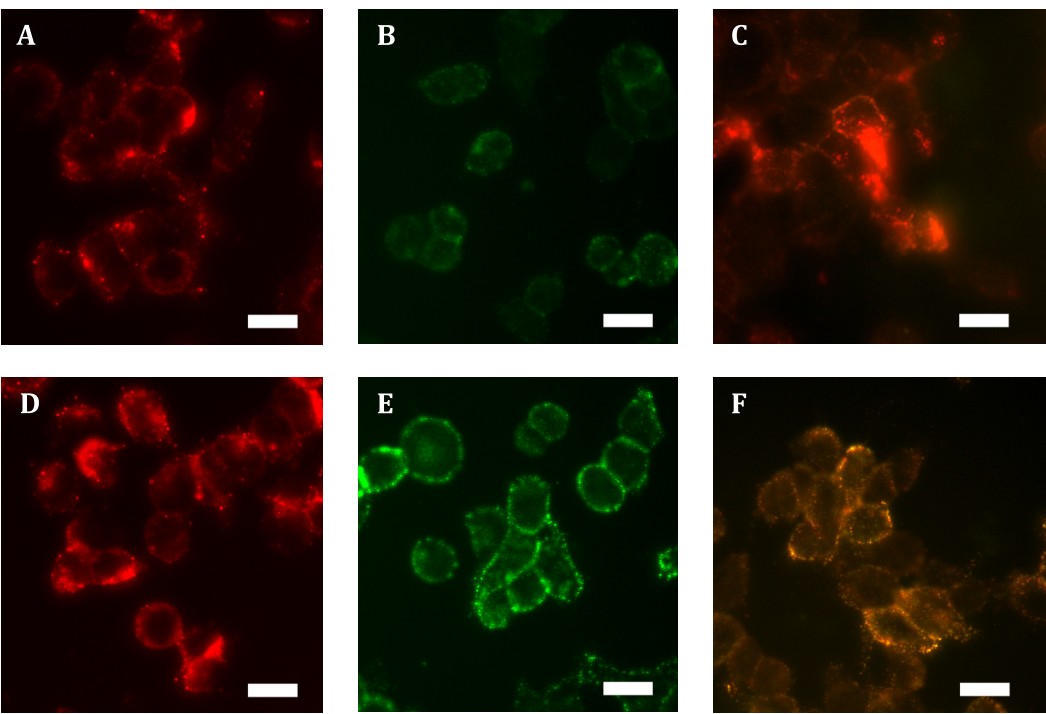

**Figure 1** **Immunocytochemistry staining of HA-tagged receptors in stably-expressing HEK cell lines.**
Anti-HA labelling of Flp-in HEK cells stably expressing HA-hGPR18 (A, B, C) or HA-TCS-hCB1 (D, E, F). (A) and (D) show the total receptor distribution in cells, while (B) and (E) show steady state surface receptors. (C) and (F) show, in green, the steady state surface receptor population (analogous to B and E) at the conclusion of a two-hour incubation with primary antibody at 37 °C, while red staining indicates the population of receptors which had traversed the cell surface at any point during this period ('cycled' receptors; secondary antibody applied under permeabilised conditions). In these images, yellow denotes co-localised staining. Scale bar, 20 μm. Representative images from at least $n = 3$.

To test whether this distribution of hGPR18 in the Flp-in HEK cell line was related to the degree of constitutive trafficking, living cells were incubated with primary anti-HA antibody for two hours, and then probed for surface and 'cycled' receptor sub-populations (which had traversed the cell surface at some point during antibody feeding, but may be intracellular at detection) by applying different colours of secondary antibody under non-permeabilising (green), then permeabilising (red) conditions, respectively. As shown in Fig. 1C, this revealed considerable intracellular cycling 'across' the cell surface, suggesting that GPR18 receptors are delivered to the cell surface but are not retained as a stable surface-resident population. As the hCB1 receptor has been shown to exhibit a substantial level of constitutive trafficking behaviour (*Grimsey et al., 2010*; *Leterrier et al., 2004*; *McDonald et al., 2007*), HA-TCS-hCB1 Flp-in HEK cells were included as a positive control over a matched period of live-feeding anti-HA antibody. As shown by Fig. 1F, the amount of receptor constitutively internalised during the live feeding period was considerably lesser than was observed for hGPR18—labelling in red is predominantly colocalised with surface receptor. This suggests that GPR18 trafficked through the cell surface at a faster rate than CB1R.

In order to generate cells with greater GPR18 expression, transient transfection was utilised. HA-hGPR18 pCAG was transiently transfected into CHO-K1 cells, and transient expression of two different HA-hGPR18 expression plasmids (pcDNA5$^{TM}$/FRT and pCAG) were compared in HEK Flp-in WT cells. Labelling of HA-hGPR18 expression by immunocytochemistry was compared to single HA-tagged hCB1R (HA-TCS-hCB1 in pcDNA5$^{TM}$/FRT).

Total receptor expression was found to differ between plasmids. Notably, HEK Flp-in WT cells transiently expressing HA-hGPR18 in the pCAG vector expressed considerably higher total amounts of receptor than both the pcDNA$^{TM}$ 5/FRT for HA-hGPR18, and the HA-TCS-hCB1 (pcDNA$^{TM}$ 5/FRT) constructs (Fig. 2A). Surface expression of the HA-hGPR18 pcDNA$^{TM}$ 5/FRT construct was substantially lower than the level of HA-TCS-hCB1 pcDNA$^{TM}$ 5/FRT expression, although the total expression level of receptor was approximately the same (Fig. 2B). CHO-K1 cells transfected with HA-hGPR18 pCAG exhibited an equivalent subcompartment expression profile to HEK Flp-in cells. Additionally, consistent with the qualitative assessment of trafficking in stably-expressing cells, GPR18 was found to undergo substantially greater constitutive surface delivery and internalisation than cells transiently expressing HA-TCS-hCB1 (Fig. 2C), and this high rate of constitutive trafficking was seen independent of the vector-associated differences in total and surface receptor expression.

These findings were verified qualitatively by widefield fluorescence microscopy which supported our observation of substantially greater 'cycled' intracellular hGPR18 (indicating constitutively internalised receptor) than HA-TCS-hCB1 in HEK Flp-in WT cells (Figs. 3A–3C). Images also allowed verification of the constitutive trafficking phenotype of GPR18 in CHO-K1 cells (Fig. 3D), suggesting this is not a phenomenon limited to HEK cells.

## cAMP assays by CAMYEL

The CAMYEL biosensor was transfected into cells, and a panel of putative GPR18 ligands was applied at high concentrations (10 μM) to detect maximum possible efficacy for inducing changes to cAMP levels (Fig. 4). Drugs included in the ligand screen were the putative GPR18 agonists NAGly, AEA and O-1602 which were all tested in the presence and absence of 10 μM forskolin (FSK); and the putative GPR18 inverse agonist AM251, which was tested in the presence and absence of 1 μM FSK to make an increase in cAMP more readily detectable. As AEA and AM251 are ligands for CB1R at the concentrations used, cAMP experiments were performed for HEK cells stably expressing 3HA-hCB1 as a control for the methodology and integrity of those ligands. In addition to the FACS sorted HA-hGPR18 Flp-in HEK stable cell line, two transient lines were also tested due to their higher cell surface expression: the HEK-FT cell line was used due to its characteristically high transfection efficiency (important as CAMYEL cAMP signalling detection is dependent on co-transfection of the biosensor and the receptor); and CHO-K1 cells were also included because GPR18 has been reported as eliciting cAMP signalling responses in this cell line (Kohno et al., 2006).
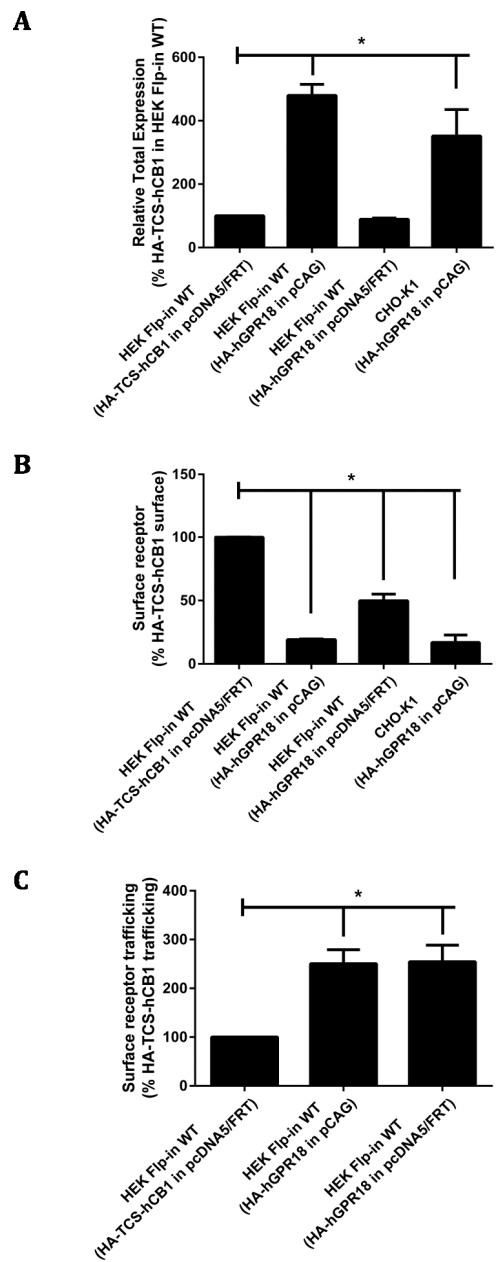

**Figure 2 Quantified receptor staining in transiently-expressing cell lines.** Immunocytochemistry was performed to selectively label the indicated receptor sub-populations (total fluorescence normalised to number of receptor-expressing cells). (A) Shows total receptor expression (primary and secondary antibodies applied under permeabilised conditions) relative to expression level of HEK Flp-in WT cells transiently expressing HA-TCS-hCB1 in pcDNA5/FRT. (B) Shows the steady-state surface receptor population (primary antibody was applied to living cells for 2 h at 37 °C, followed by secondary antibody being applied to the live cells when chilled on ice), normalised for total receptor expression level (A), and relative to the surface expression level of HA-TCS-hCB1 in HEK Flp-in WT cells. (C) Represents the extent of constitutive receptor internalisation over the 2 h course of primary antibody live feeding, by measuring the total 'cycled' receptor (secondary antibody applied to permeabilised cells) normalised to steady-state surface expression (B), and is expressed relative to the degree of cycling of HA-TCS-hCB1 transiently expressed in HEK Flp-in cells. (C) Therefore shows how much of the steady-state surface receptor expression level was constitutively trafficked via the cell surface in 2 h relative to HA-TCS-hCB1 ($n = 3$).

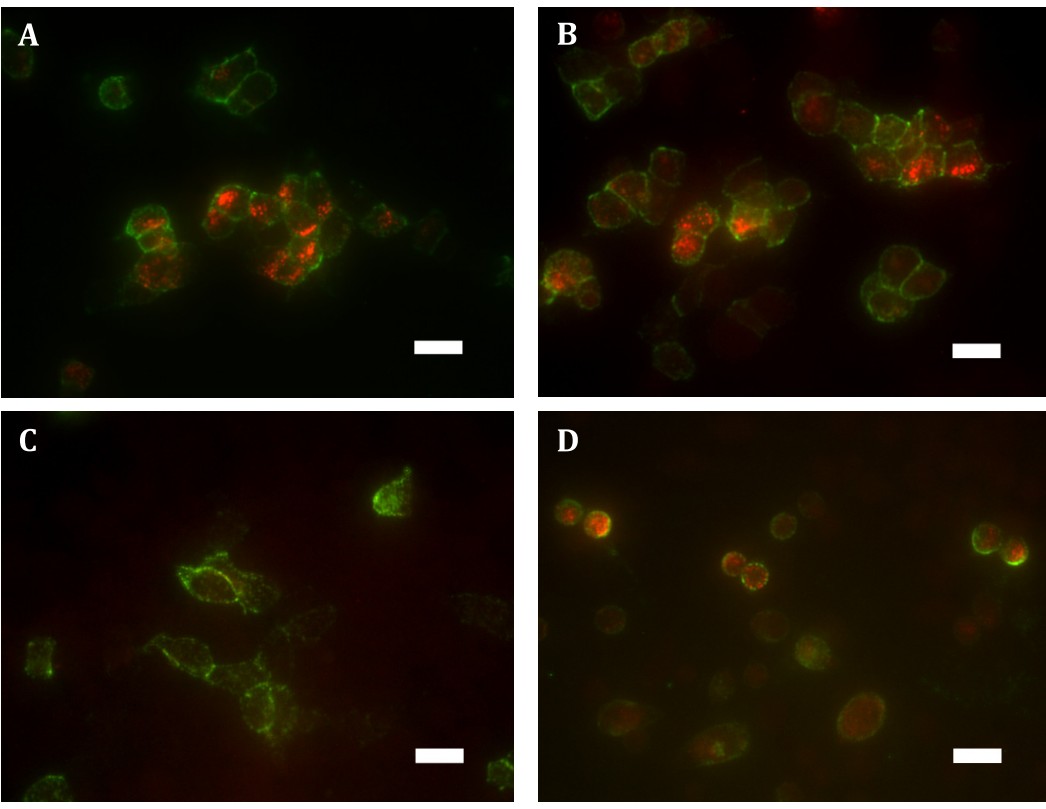

**Figure 3** **Immunocytochemistry staining of HA-tagged receptors in transiently-expressing cells.** Labelling of HEK Flp-in cells transiently expressing HA-hGPR18 pCAG (A), HA-hGPR18 pcDNA5/FRT (B), or HA-TCS-hCB1 pcDNA5/FRT (C); and CHO-K1 cells transiently expressing HA-hGPR18 pCAG (D). Green staining distinguishes the steady-state surface receptor population, while red staining shows the population of receptors that had traversed the cell surface at any point during a two-hour primary antibody 'live feeding' period ('cycled' receptors; secondary antibody applied under permeabilised conditions). Scale bar, 20 $\mu$m. Representative images from $n = 3$.

The 3HA-hCB1 HEK cell line elicited the expected Gi-linked cAMP signalling phenotype on stimulation with CB1R agonists AEA and CP55,940 (Fig. 4A). The inverse agonist AM251 resulted in an increase in cAMP levels above those produced by FSK alone in 3HA-hCB1, in keeping with a reduction in constitutive Gi-linked signalling of these receptors. In stable HA-hGPR18 Flp-in HEK cells, the only significant ligand-mediated effects were modest increases in cAMP induced by of AM251 and O-1602 (Fig. 4B). This AM251 effect was also observed in the transiently-expressing HEK-FT cells (Fig. 4C). To determine if this was GPR18 mediated, AM251 was tested in HEK Flp-in WT cells not transfected with receptor and was also found to increase cAMP to approximately the same extent, thus this effect was not specific to GPR18. Finally, no significant ligand-mediated changes to cAMP were detected in CHO-K1 cells transiently transfected with hGPR18.

## Phospho-ERK assays

Similar to the cAMP screening approach described above, assays for pERK were designed to elicit Emax responses measured over a time course, for a panel of putative GPR18 ligands.

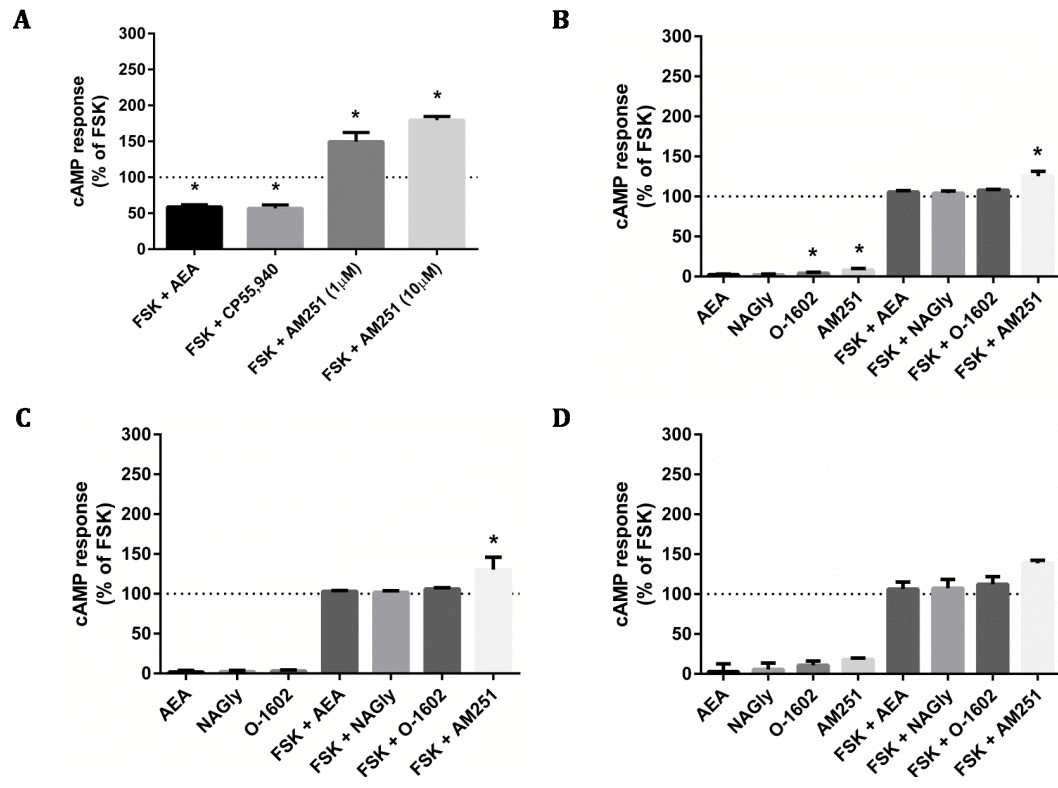

**Figure 4** **cAMP signalling (CAMYEL) in stably and transiently receptor-expressing cells.** Area-under-the-curve summary data for cAMP signalling in real-time-detection system, over a time course of approximately 25 min. Data are normalised for vehicle (0%) and matched FSK (100% = 10 μM FSK for all responses except AM251, for which 1 μM FSK was applied). 3HA-hCB1 HEK cells (A) were treated with 10 μM of cannabinoid agonists CP55,940 and AEA, and with 1 and 10 μM of the inverse agonist AM251. (B) Shows cAMP responses in the HA-hGPR18 Flp-in HEK cell line with 10 μM of NAGly, O-1602, AM251 and AEA. (C) Shows cAMP responses in HEK FT cells transiently expressing HA-hGPR18 (pCAG), and (D) shows signalling in CHO-K1 cells transiently expressing HA-hGPR18 pCAG. ($n = 2$–5).

For methodological validation and to confirm (where possible) the integrity of ligands, a HEK cell line stably expressing 3HA-hCB1R was also screened for ligand-mediated pERK stimulation.

As expected, stimulation of the 3HA-hCB1R with 10 μM AEA elicited a robust, highly transient pERK response that peaked at approximately 4 min (Fig. 5A). When analysed by area-under-the-curve, the magnitude of this pERK stimulation peak was significantly different from vehicle (Fig. 5B). In contrast, the panel of putative GPR18 ligands (10 μM) failed to alter the basal level of pERK at any point in stimulation time courses on the HA-hGPR18 Flp-in HEK cell line (Figs. 5C–5D) or HEK Flp-in cells transiently transfected with HA-hGPR18 in pCAG (Figs. 5E–5F). This line was selected instead of the HEK-FT line because of its lower basal level of pERK and more distributed (as opposed to clustered) cell growth pattern, which simplified interpretation and analysis of high-content immunocytochemistry data. High content analysis of pERK immunocytochemistry enabled quantitative analysis of transfected or untransfected cells within the same field of view. No putative GPR18 ligand induced a detectable pERK response either in cells expressing

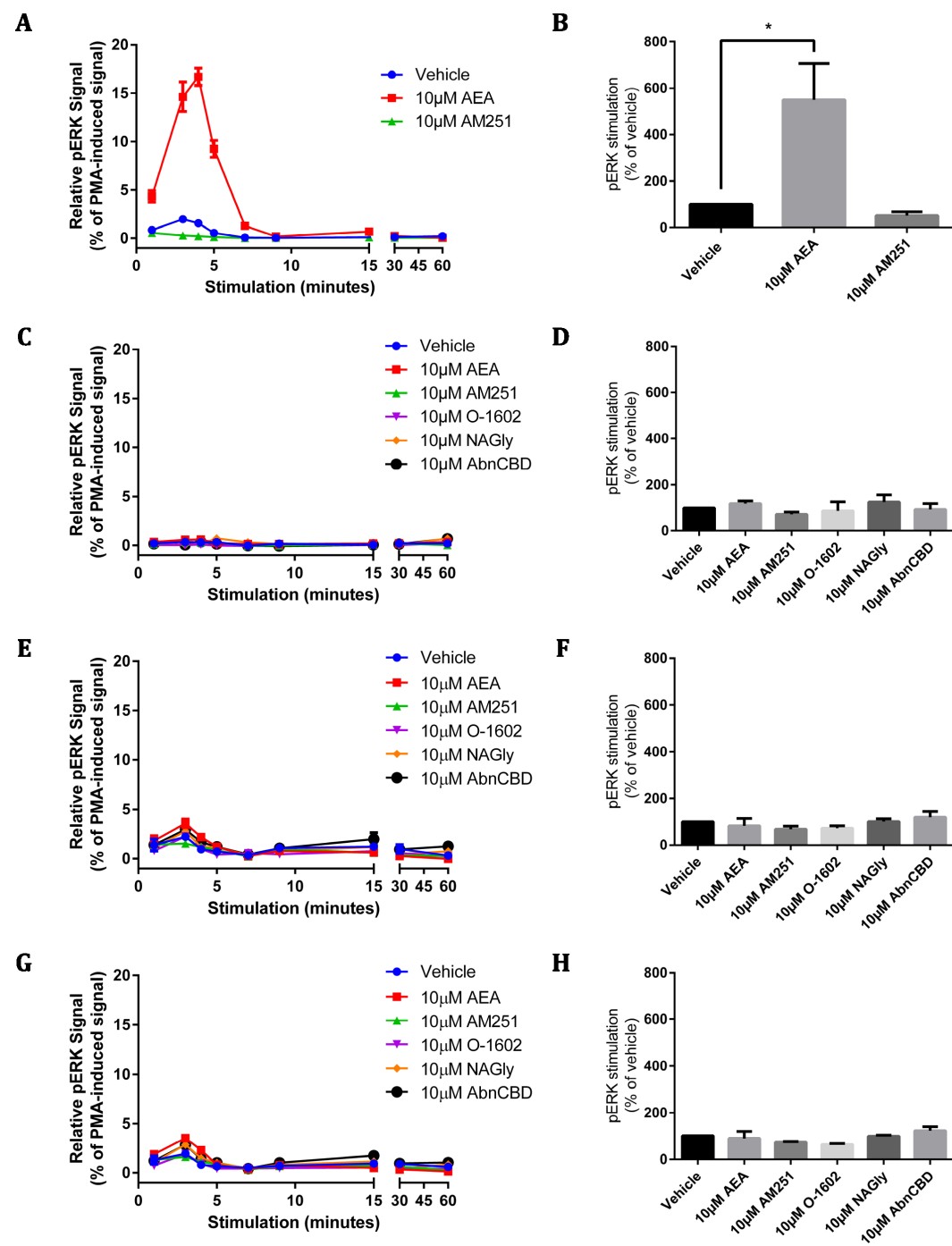

**Figure 5 pERK signalling responses in stably and transiently receptor-expressing HEK cells.** Time courses (A, C, E, G) are representative data from single experiments, normalised for pERK level induced by treatment with 100 nM PMA for 4 min 30 s (100%) and 10 μM U0126 for 30 min (0%). Area-under-the-curve analysis (B, D, F, H) is combined data from three biological replicates (B and D) or two biological replicates (F and H). (A) and (B) show pERK stimulation of HEK cells stably expressing 3HA-hCB1 receptors, (C) and (D) show pERK stimulation of HEK Flp-in cells stably expressing HA-hGPR18, (E) and (F) show pERK stimulation of HEK Flp-in WT cells transiently expressing HA-hGPR18 pCAG (receptor-expressing subpopulation only), and (G) and (H) show pERK stimulation of the same HEK Flp-in WT cells, but for the subset of cells not expressing receptor.

(Figs. 5E–5F) or not expressing receptor (Figs. 5G–5H). A low magnitude vehicle response was seen at the 3 min time point, but this was also present in the sub-population of cells that did not express receptor, and was therefore not receptor-mediated.

## pplss-3HA-hGPR18 Flp-in HEK cells

As no detectable ligand mediated responses could be identified by any of the assays utilised in either stably or transiently transfected cell lines, we made a final attempt to produce a cell line with substantially higher cell surface expression by generating a GPR18 construct N-terminally tagged with a preprolactin signal sequence and three HA-tags (pplss-3HA-hGPR18 in pcDNA$^{TM}$5/FRT), which was then stably expressed in HEK Flp-in cells. Again, the Flp-in system failed to generate a cell line with homogeneous receptor expression, with some Hygromycin B-resistant cells expressing very high surface receptor levels, but others below the limit of detection. Nonetheless, in cells expressing detectable GPR18, surface expression was found to be equivalent to the 3HA-hCB1 HEK line (a line known to have adequate surface receptor expression to elicit robust signalling responses; Fig. 6A). Assessment of total receptor expression revealed that the ratio of intracellular to surface-expressed receptor was similar to the first Flp-in HEK cell line used in this study (HA-hGPR18 Flp-in HEK), although the pplss-3HA-hGPR18 cells expressed vastly greater amounts of receptor per cell.

This cell line was then used to detect pERK following stimulation with a range of putative GPR18 ligands. No statistically significant differences in pERK levels were found between ligand conditions, nor between cells that co-stained for GPR18 and cells that did not co-stain for receptor (Figs. 6B and 6C).

As ligand-mediated responses had not been detected in any assays thus far, we queried whether valid pathways were being studied. A common functional consequence of GPCR binding and activation by a cognate agonist is that it responds by undergoing desensitisation followed by rapid internalisation. Trafficking assays were therefore performed on the pplss-3HA-hGPR18 Flp-in HEK cell line. None of the ligands tested elicited significant alterations to hGPR18 surface expression in pplss-3HA-hGPR18 Flp-in HEK cells with 15 min (Fig. 7A), 1 h (7C), or 18 h (7E) of stimulation. 3HA-hCB1 receptors internalised and were degraded in response to AEA treatment as expected (Figs. 7B, 7D, 7F and 7H).

A constitutive trafficking phenotype of GPR18 was described above, utilising a primary antibody live-feeding approach to show that GPR18 undergoes considerably greater surface membrane turnover than CB1R. A previous GPR18 study utilising histidine auxotrophic yeast also postulated that the receptor is constitutively active (Qin et al., 2011), and that this activity could be diminished by a substitution mutation of residue 108 (A108N) of the hGPR18 gene. We expressed HA-hGPR18 A108N transiently in HEK Flp-in WT cells in parallel with the standard HA-hGPR18 pcDNA$^{TM}$5/FRT plasmid. After normalising to total expression, the GPR18 A108N mutant receptor had a higher proportion of stable surface expression than WT hGPR18 (Fig. 8A). In case this putatively less constitutively active GPR18 mutant receptor responded to agonist, pERK assays were performed. However, 10 $\mu$M NAGly produced no alteration in pERK activation, although a control condition

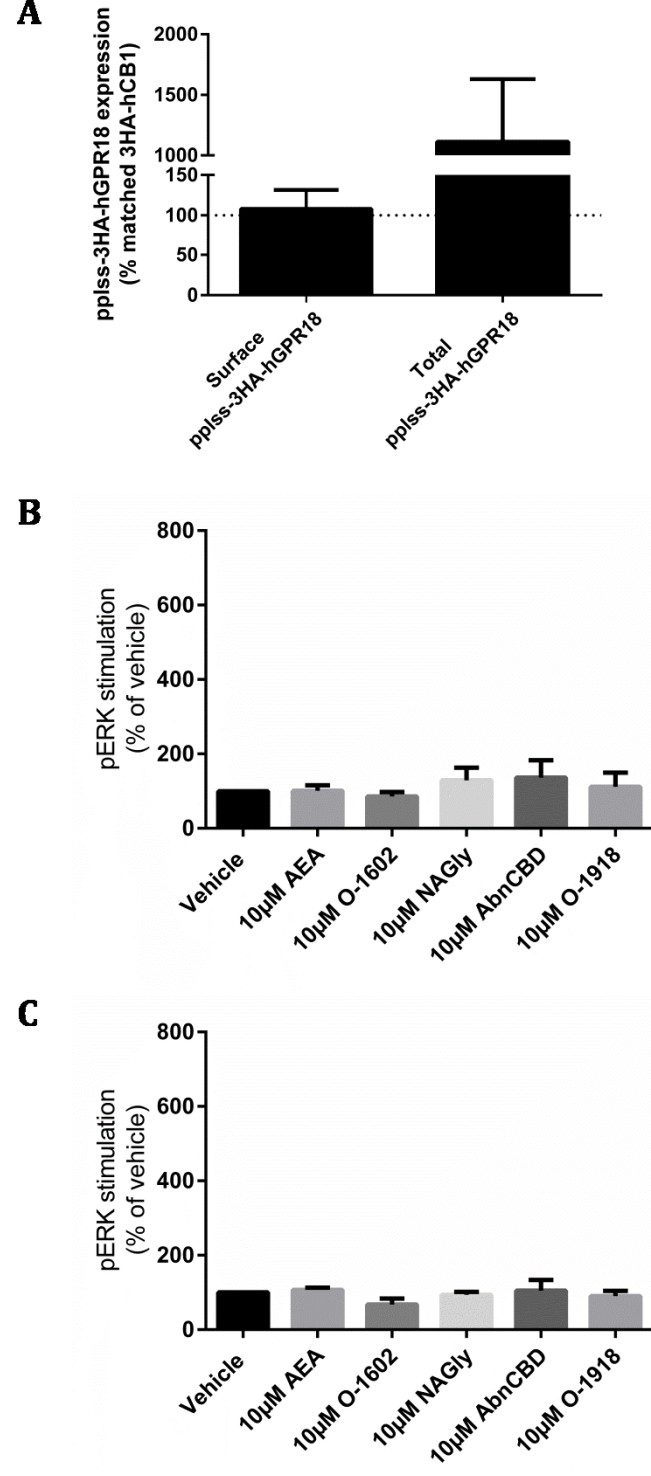

**Figure 6 pplss-3HA-hGPR18 HEK Flp-in cell expression and signalling.** (A) Is combined data from three biological replicates, comparing surface and total receptor expression in the pplss-3HA-hGPR18 Flp-in HEK cell line relative to the 3HA-hCB1 HEK stable cell line. Summary data (B, C) is area-under-the-curve analysis performed for combined data from three biological replicates. (B) shows pERK stimulation in pplss-3HA-hGPR18 Flp-in cells positive for GPR18 expression (based on anti-HA co-labelling), while (C) shows pERK stimulation in the same experiment but from cells that do not stain for receptor.

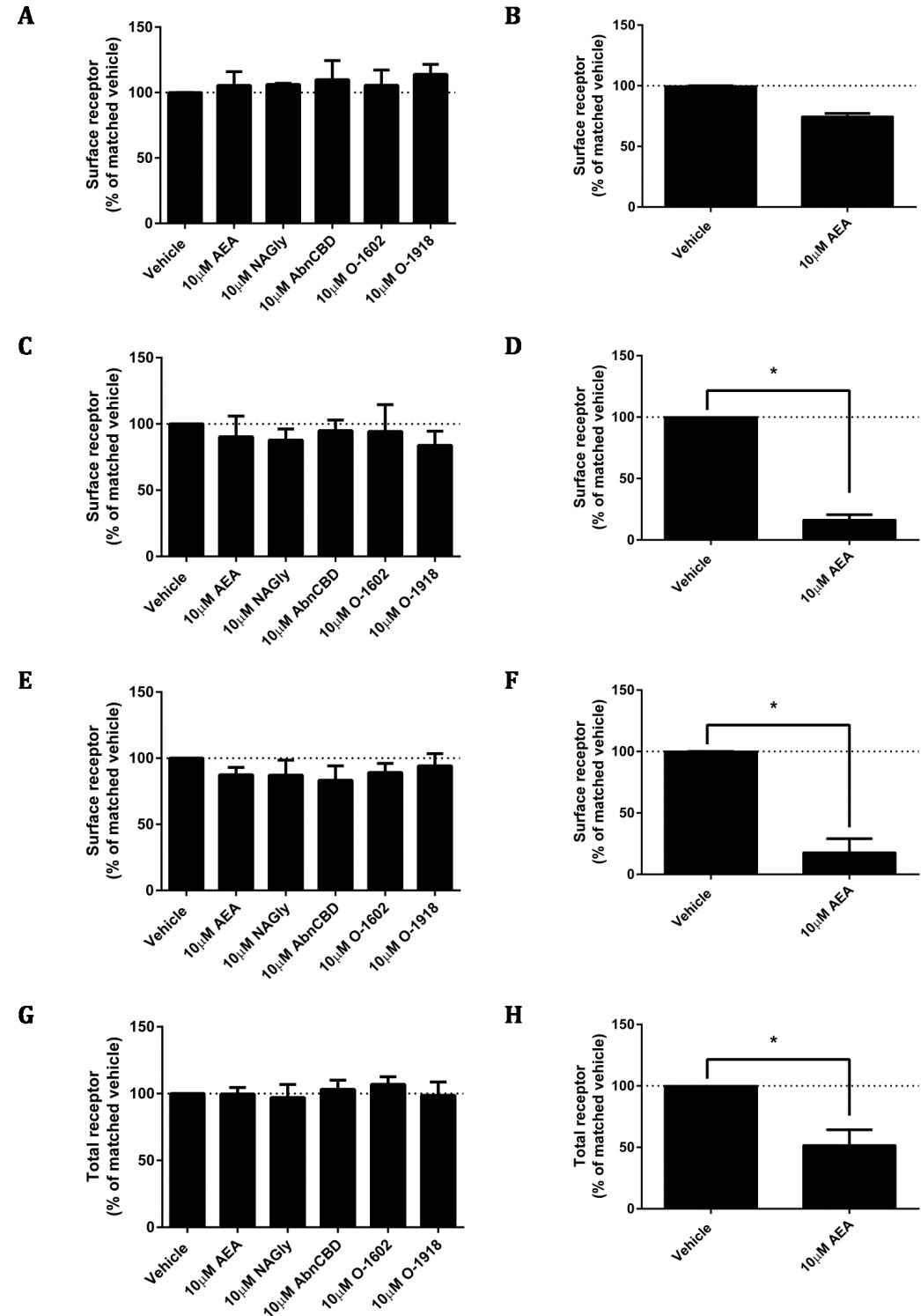

**Figure 7 Ligand-induced changes to receptor localisation.** Changes to receptor surface expression (in comparison with vehicle) following stimulation with high concentrations of putative ligands, at time points of 15 min (A, B), 1 h (C, D) and 18 h (E, F) for HEK Flp-in cells stably expressing pplss-3HA-hGPR18 (A, C, E, G) and HEK cells stably expressing 3HA-hCB1 (B, D, F, H). (G and H) show changes in total expression (in comparison with vehicle) at 18 h ($n = 3$).

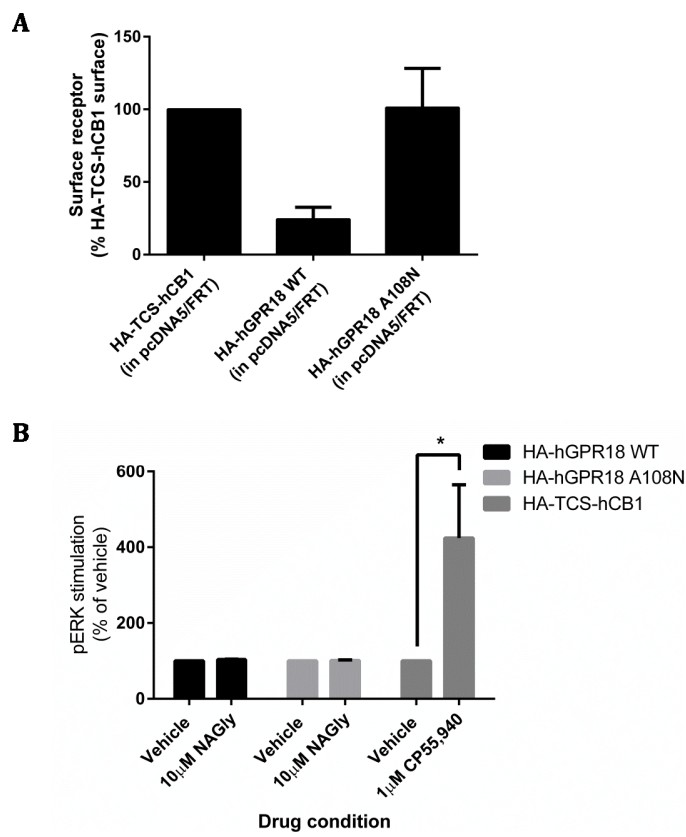

**Figure 8**  **GPR18 phenotype modifications by A108N point mutation.** As for the data in Fig. 2, immuno-cytochemistry was performed to selectively label the indicated receptor sub-populations. (A) Shows the steady-state surface receptor population (primary antibody was applied to living cells for 2 h at 37 °C, followed by secondary antibody being applied to the live cells when chilled on ice), normalised for total receptor expression level, and presented relative to the surface expression level of HA-TCS-hCB1 in HEK Flp-in WT cells ($n = 4$). (B) Shows the area-under-the-curve for pERK stimulation of hGPR18 WT, hGPR18 A108N mutant, and HA-TCS-hCB1 receptors transiently expressed in HEK Flp-in WT cells ($n = 3$).

(transiently expressed HA-TCS-hCB1 in HEK Flp-in WT, stimulated with 1 μM CP55,940) once again validated assay performance (Fig. 8B).

## Endogenous expressers of GPR18

The disappointing inability to reproduce published studies on HEK cells expressing hGPR18 led us to search for cells that endogenously express GPR18, based on the hypothesis that endogenous expressers should express the appropriate cellular machinery for mediation of receptor function. Previous reports have suggested that the mouse microglial BV-2 cell line, and human endometrial cell line HEC-1b both express functional GPR18 (*McHugh et al., 2012a*; *McHugh et al., 2014*; *McHugh et al., 2012b*). We therefore screened BV-2 cells (*Gibbons, Sato & Dragunow, 2003*) and HEC-1a cells (the HEC-1b parental line) by reverse-transcription PCR. However, neither of these cell lines expressed detectable GPR18 mRNA (data not shown). Reports of glial expression (*McHugh et al., 2010*) of GPR18 led us to screen local provenance primary human GBM cell lines, developed by

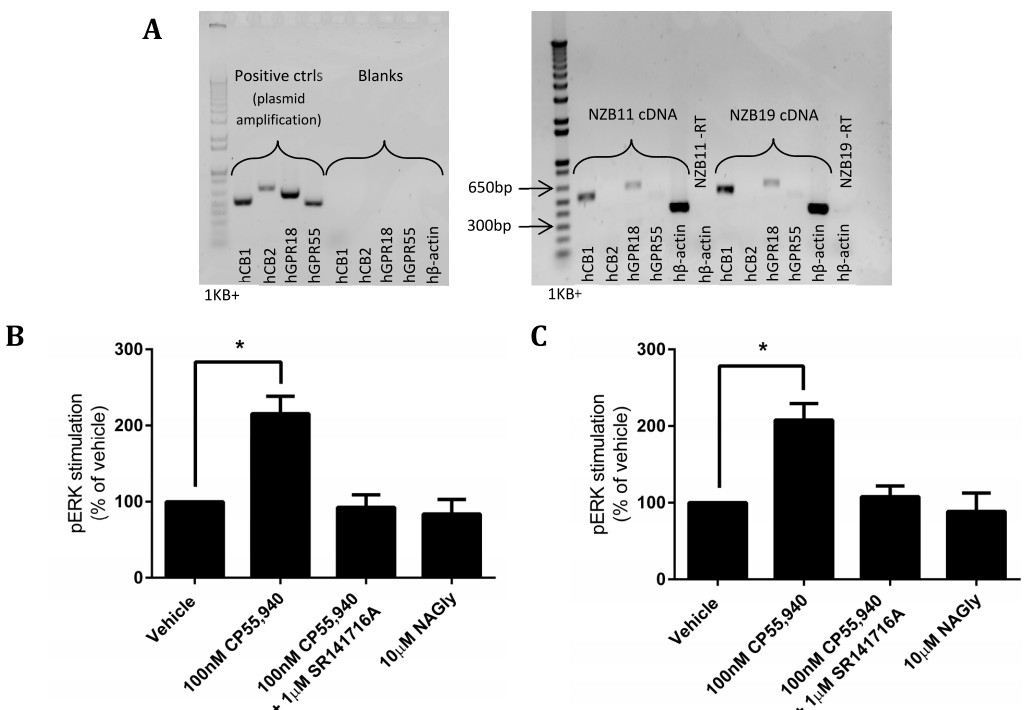

**Figure 9 mRNA expression and pERK stimulation in two endogenously GPR18-expressing GBM cell lines.** (A) DNA gel image of PCR products showing the mRNA expression profile of genes of interest in the GBM lines NZB11 and NZB19. Plasmid templates were utilised as positive size-reference controls, and reactions without template ("blanks") as negative controls. Primer sets are shown across the bottom of the gels. Representative of three independent experiments. (B–C) pERK detection of single time point (5 min) stimulations of NZB11 (B) and NZB19 (C). (B) and (C) are combined data from three independent biological replicates, each normalised to vehicle.

the Auckland Cancer Society Research Centre. NZB11 and NZB19 are cell lines from the NZB bank of brain cancer cell lines, and are typed as being of right-parietal and right temporal origin respectively. Both lines were found by RT-PCR to express hCB1 and hGPR18 mRNA, but not hCB2 or hGPR55 mRNA (Fig. 9A). In both lines, pERK was stimulated on treatment with 100 nM CP55,940 in a manner sensitive to 1 μM SR141716A (emphasising the specificity of this response to CB1R), but no response was detected on treatment with 10 μM NAGly in either cell line (Figs. 9B and 9C).

## DISCUSSION AND CONCLUSIONS

Several studies indicate that when GPR18 is stimulated with cannabinoid ligands, including NAGly, functional consequences ensue (*Kohno et al., 2006*; *McHugh et al., 2010*; *McHugh et al., 2012a*; *McHugh et al., 2012b*). However, publications by *Yin et al. (2009)* and *Lu, Puhl & Ikeda (2013)* disclose comparatively anomalous findings as to the responsiveness of GPR18. This study aimed to clarify the function of GPR18 in response to putative ligands.

We report that GPR18 is a challenging receptor to express heterologously, with high intracellular receptor expression and variable overall expression even when utilising the Flp-in system, which would be expected to result in an isogenic cell population with

homogeneous expression (as we have been able to achieve with other receptors). Despite low cell surface expression, the high level of total expression in some cells, combined with the previous suggestion that the receptor exhibits high levels of constitutive signalling activity (*Qin et al., 2011*), led us to examine whether the receptor was efficiently delivered to the cell surface but rapidly internalised. This led to the first notable finding of our study, wherein antibody "feeding" experiments suggested that wildtype GPR18 exhibited an unusually rapid rate of constitutive internalisation from the cell surface. We also observed that, in line with the previous report by *Qin et al. (2011)*, introduction of an A108N mutation reduced this constitutively cycling phenotype and resulted in higher steady-state surface receptor expression. Considering that steady-state surface expression of the wildtype receptor is extremely low, but the quantity of receptor that is delivered to and internalised from the cell surface over a given period is very high, the magnitude of agonist-mediated receptor signalling responses may be reduced (or wholly absent) because cell surface receptor is not available to interact with ligands. A highly constitutively active phenotype may also argue against the need for an endogenous receptor agonist, instead implying that an endogenous ligand might be more likely to influence signalling via inverse-agonism. However, there was no evidence in our signalling assays for high rates of basal activity in cells expressing GPR18 that would support a high level of activity to correspond with this high rate of trafficking.

*Kohno et al. (2006)* reported the ability of NAGly to inhibit cAMP in CHO-K1 cells stably expressing hGPR18. Therefore we investigated the ability of a panel of putative agonists and inverse agonists, to modulate cAMP. The assay was validated by the stimulation of significant signalling responses in HEK cells expressing 3HA-hCB1R, which signalled robustly and as expected (see Fig. 4). When treated with agonists (AEA or CP55,940), hCB1 receptor-expressing cells signalled in a G$\alpha$ i/o-associated manner to reduce FSK-induced cAMP levels.

In conflict with the data from *Kohno et al. (2006)*, G$\alpha$i/o responses (reduction in FSK-induced cAMP levels) were not detected on treatment of cells stably- or transiently-expressing hGPR18 with a high concentration of NAGly, nor with any of the other putative agonists included in the screen. Small but significant increases in cAMP levels were found to be induced by high concentrations of the putative GPR18 inverse agonist AM251 in some GPR18-expressing cells, however this observation was determined not to be a GPR18-mediated effect.

GPR18-mediated phosphorylation of ERK has been reported twice for HEK cell models of GPR18 (*McHugh et al., 2010*; *McHugh et al., 2012a*), with concentration-dependent induction by a large number of cannabinoid ligands, including NAGly (EC$_{50}$ 44.5 nM), O-1602 (65.3 nM), and AEA (3.8 $\mu$M) (*McHugh et al., 2012a*). However, in our study no significant ligand-mediated responses were detected in pERK assays, in any of the stable or transiently expressing GPR18 cell models utilised (Figs. 5 and 6). In contrast, robust pERK activation was observed in the HEK-hCB1 cells confirming the integrity of AEA utilised in this study, and the ability to detect receptor-mediated changes in pERK in our assay design and in the HEK cell background.

While it is feasible that inclusion of an HA-tag at the N-terminus of GPR18 may have influenced secretion and/or trafficking of the receptor, this seems unlikely given the wide

utilisation of such tags in GPR18 research and reports of analogous trafficking and signalling to untagged receptors (*Grimsey et al., 2010*; *Ivic et al., 2002*; *Mason, Kozell & Neve, 2002*). Furthermore, inclusion of an HA tag does not explain the discrepancy between our findings and prior reports of GPR18 signalling as our construct is very similar to the HA-tagged construct utilised by *McHugh et al. (2010)* and *McHugh et al. (2012a)*.

Even though transient transfection resulted in high levels of receptor expression the possibility remained that the lack of signalling detected in these assays was due to insufficient receptor expression. Thus, we generated a HEK cell line which encoded hGPR18 with the pplss signal peptide. This cell line showed greatly increased cell surface expression (equivalent to our 3HA-hCB1 HEK cell line), and overall expression well in excess of the 3HA-hCB1 line, reflecting the previously observed pattern of a low relative surface:total expression ratio. Empirically, the fold change in expression level produced by pplss tagging was approximately one order of magnitude, although a more precise estimation of the consequence of pplss tagging on total expression level was not possible due to the original single HA-tagged hGPR18 Flp-in cell line anti-HA staining intensity being at the lower limit of detection. Despite the substantial surface expression of receptors, this cell line also failed to show a detectable pERK response upon exposure to a range of putative GPR18 ligands.

A lack of response in cAMP or pERK assays could indicate that in our cells these are not pathways to which GPR18 is linked. While Gαi-mediated responses were expected on the basis of previous reports (e.g., *Kohno et al., 2006*), these assays would also have detected Gαs-mediated increases in cAMP, and Gi, Gs or Gαq linked alterations in pERK. However, we aimed to look at a signalling pathway that wouldn't require presupposition of any specific signalling pathway activation by looking at internalisation of cell surface receptors. GPCRs typically undergo rapid internalisation in response to agonist stimulation (reviewed in *Drake, Shenoy & Lefkowitz, 2006*). Therefore, we examined whether ligand-induced changes in cell surface receptor expression could be detected in the pplss-3HA-hGPR18 HEK cells. As for the other assays described, hCB1R-expressing cells were utilised as a positive control for the efficacy of the ligands known to be also active at CB1R and to validate assay design. As expected, hCB1R rapidly (within one hour) internalised on treatment with AEA. In pplss-3HA-hGPR18-expressing cells, however, agonist-mediated effects were not significantly induced at 15 min, 1 h or 18 h.

The observed stark lack of ligand-mediated hGPR18 activity was troubling in light of the published data, therefore we turned to endogenously expressing cell lines. BV-2 cells have been reported both as endogenous GPR18 expressers (*McHugh et al., 2014*; *McHugh et al., 2012b*) and non-expressers (*Atwood et al., 2011*), while HEC-1b endometrial cells have also been reported to express GPR18. We therefore screened the HEC1b parental cell line, HEC-1a, and BV-2 cells in our laboratory by PCR. Unfortunately, no GPR18 was detected in either cell line. As GPR18 had been suggested to be expressed in glial cells we screened a series of low passage human glioblastoma cell lines for GPR18 expression. The first two cell lines characterised expressed mRNA for GPR18 and CB1, but not CB2 or GPR55 (expression of CB2R or GPR55 could have confounded results due to the reported potential of both to respond to some of the same ligands as GPR18 and CB1R, such as AEA *Ryberg et al., 2007*). We therefore examined the ability of NAGly and the CB1R agonist

CP55,940 to stimulate pERK response in these cells. While a robust response was observed to CP55,940 that could be blocked by CB1R-selective concentrations of SR141716A, no response was observed to NAGly in these cells.

Thus, these experiments were unable to reproduce the results of earlier studies that suggested that NAGly and other cannabinoid ligands can activate classical G$\alpha$i/o signalling pathways in GPR18 expressing cells (*Kohno et al., 2006*; *McHugh et al., 2012a*). While low cell surface receptor expression may explain this for the HA-hGPR18 Flp-in HEK cells, the transiently transfected hGPR18 lines and pplss-3HA-hGPR18 cells had surface receptor levels equivalent to those observed for the 3HA-hCB1 cell line, for which G$\alpha$i/o-coupled signalling responses were readily detected. Given previous signalling was carried out in both HEK and CHO-K1 cells, the lack of signalling observed here is hard to explain on the basis of cell background, although it is clear that cell lines within different laboratories can show considerable phenotype drift (as evidenced by the lack of agreement of GPR18 expression in BV-2 cells). Our studies thus support the work of *Lu, Puhl & Ikeda (2013)* who were likewise unable to demonstrate activation of canonical GPCR pathways by GPR18, and *Yin et al. (2009)*, who were unable to detect arrestin recruitment. It might be tempting to speculate that lack of stability of NAGly accounts for the differences between laboratories. However, anandamide has also been shown to act as a GPR18 agonist, and due to its well-characterised agonist classification at hCB1R this ligand constituted a useful control, demonstrating AEA integrity in all our experiments.

While it may be (and our data would support) that the endogenous ligand for GPR18 has not yet been identified, it is also possible that GPR18 might only be functional when co-expressed with another protein, or perhaps acts as a decoy receptor with the ability to recognise ligands with high affinity and specificity, but not usually capable of activating downstream signalling pathways. A decoy receptor may sequester ligands and reduce the concentration available to activate other canonically signalling receptors (*Mantovani, Bonecchi & Locati, 2006*; *Mantovani et al., 2001*). The possible role of GPR18 as a decoy receptor may appear to be incongruent with much of the current literature due to evidence for GPR18 signalling, particularly through pathways leading to ERK phosphorylation as previously described (*McHugh et al., 2010*; *McHugh et al., 2012a*). However, this characterisation need not preclude GPR18 from being assigned to this group of receptors; for example, decoy receptor CCX-CKR recruits $\beta$-arrestins (*Watts et al., 2013*), which have been shown (for other receptors) to activate signalling cascades leading to ERK phosphorylation (*Gudermann, 2001*; *Shenoy et al., 2006*).

The premise that some GPCRs require a 'co-receptor' or interacting partner to function is well established for GPCRs. For example, Wnt receptors, require the presence of a co-receptor such as lipoprotein receptor-related protein 6 (LRP6) in order to signal (*Komiya & Habas, 2008*). If GPR18 does require a co-receptor, this (combined with potential differences in characteristics of cell lines between laboratories), may explain why only certain laboratories observe ligand-elicited GPR18 signalling responses. However, our inability to detect a response in endogenously expressing human glioblastoma cell lines might suggest this is unlikely.

Given that all the studies on GPR18 signalling to date appear to have been carefully carried out and well controlled, the differences between them remain puzzling. Until the cause of these discrepancies is determined, deorphanisation of GPR18 remains elusive.

## ACKNOWLEDGEMENTS

We gratefully acknowledge Associate Professor Heather Bradshaw (Indiana University) for donating the HA-hGPR18 pCAG plasmid, Mr Stephen Edgar (University of Auckland) for technical assistance with FACS, Distinguished Professor Bruce Baguley (University of Auckland) for allowing access to the GBM cells, Associate Professor Kevin Pfleger (Harry Perkins Institute of Medical Research) for providing the HEK-FT cell line, and Professor Larry Chamley (University of Auckland) for providing the HEC-1a cell line.

### Funding

Funding was received from the Maurice and Phyllis Paykel Trust (grant number 3702645 (9132)) and the School of Medical Sciences, University of Auckland. The funders had no role in study design, data collection and analysis, decision to publish, or preparation of the manuscript.

### Grant Disclosures

The following grant information was disclosed by the authors:
Maurice and Phyllis Paykel Trust: 3702645 (9132).
School of Medical Sciences, University of Auckland.

### Competing Interests

The authors declare there are no competing interests.

### Author Contributions

- David B. Finlay conceived and designed the experiments, performed the experiments, analyzed the data, wrote the paper, prepared figures and/or tables, reviewed drafts of the paper.
- Wayne R. Joseph contributed reagents/materials/analysis tools, wrote the paper, reviewed drafts of the paper.
- Natasha L. Grimsey conceived and designed the experiments, analyzed the data, contributed reagents/materials/analysis tools, wrote the paper, prepared figures and/or tables, reviewed drafts of the paper.
- Michelle Glass conceived and designed the experiments, analyzed the data, contributed reagents/materials/analysis tools, wrote the paper, reviewed drafts of the paper.

### Human Ethics

The following information was supplied relating to ethical approvals (i.e., approving body and any reference numbers):
Health and Disability Ethics Committees Ethics Reference AKL/2000/AM03.

## Ethics

The following information was supplied relating to ethical approvals (i.e., approving body and any reference numbers):

University of Auckland Biological Safety Committee (under the New Zealand Environmental Protection Authority) approval number B00343-GMO08-UA003 (s67a).

## Data Availability

Raw data is available as Supplemental Information.

## Supplemental Information

Supplemental information for this article can be found online at http://dx.doi.org/10.7717/peerj.1835#supplemental-information.

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
