# Peer review of "GPR18 undergoes a high degree of constitutive trafficking but is unresponsive to N-Arachidonoyl Glycine"

_PeerJ, doi:10.7717/peerj.1835_

## Round 0.1 · original submission · Major Revisions

Please consider the comments and concerns of the expert reviewers in your revised manuscript.

Reviewer 1 ·

Basic reporting

No comments

Experimental design

No comments

Validity of the findings

No comments (see below)

Additional comments

The findings of the present study contradict earlier findings that NAGly activation of GPR18 induces ERK phosphorylation and inhibits cAMP thereby arguing against NAGly as being the endogenous GPR18 agonist. Further, the authors were not able to replicate reported findings on GPR18 expression in some cell lines. However, a novel finding of the present study is the ability of GPR18 to undergo constitutive receptor membrane trafficking. While interesting, the biological relevance of this new finding and how such trafficking is endogenously triggered have not been elucidated. Further, as detailed below, there are some additional concerns about the present study.
Concerns:
1. The authors used equivalent concentrations of all ligands in their study. Did the authors conduct concentration response studies for NAGly and ABN-CBD?

2. The immunocytochemistry, a semi quantification method, was used to measurement of pERK. Did author run western blot to quantify the pERK level?

3. The abstract and the results sections are too long, and difficult to follow.

Reviewer 2 ·

Basic reporting

The authors demonstrated the high degree of constitutive trafficking and the lack of ligand mediated response of GPR18. The phenomenon is interesting and unique, and methodology is well organized.

Experimental design

No comments

Validity of the findings

No comments

Additional comments

There are some points however need to be clarified.

Comment#1. In figure 1-3, the authors utilized the hCB1 fused with single HA and TCS. On the other hand, in figure 4 and later, the authors utilized the hCB fused with 3HA. Is the intracellular distribution of the hCB fused with 3HA is same as the hCB1 fused with single HA and TCS?

Comment#2. As authors mentioned about the signal sequence of the pre-prolactin on the line 171, signal sequence which usually located at the N-terminal of the protein may be important for the secretion efficiency. Doesn’t HA-tag added on N-terminal of the hGPR18 affect the surface expression? And, if the signal sequence of GPR18 is cleaved after surface expression like pplss or typical signal peptides, does the complete structure of HA-hGPR18 remain intact in the cell surface?

Comment#3. In addition to Comment#2, to clarify the high degree of constitutive trafficking of GPR18, the result utilizing GPR18 which is expressed endogenously is needed.

Comment#4. In figure 6A, surface and total receptor expression of pplss-3HA-hGPR18 should be compared with 3HA-hGPR18 in addition with 3HA-hCB1.

Comment#5. Since the result of hCB1 and hGPR18 expression in NZB11 or NZB19 cells are important, it should be shown in figure 9.

Comment#6. Why did the author particularly mention about the lack of GPR55 in NZB11 and NZB19 on the line 597?

---

## Round 0.2 · Major Revisions

Your manuscript has been re-reviewed by the 2 original reviews and both have suggested additional edits. The manuscript has been improved although some new modifications are still needed.

Reviewer 1 ·

Basic reporting

No Comments

Experimental design

No Comments

Validity of the findings

No Comments

Additional comments

The authors have provided reasonable responses to the reviewers' comments, and made reasonable effort to revise their manuscript.

There is clear need to correct a number of typographical errors in the revised manuscript. For example, utilization not "utilisation" should be used in the first paragraph on page 15. Also, on page 15 CB2 should be CB2R and GPr18 should be GPR18. Therefore, there is a clear need to thoroughly proofread the revised manuscript.

Reviewer 2 ·

Basic reporting

NA

Experimental design

NA

Validity of the findings

NA

Additional comments

To the response for comment #1

Even the authors changed from 1HA to 3HA on hCB1 according to the appropriate control for the hGPR18 constructs possessing the number of the HA tag, there may be a reason why you change the number of HA tag on these constructs. This reason should be noted; otherwise it may confuse us about what is the difference between these constructs.

To the response for comment #2

Actually, I think it is the most important point that phosphorylation of ERK by NAGly was investigated in previous paper, but not in present paper, even using the similar plasmid and cells. Therefore, first I wonder whether the integrity of receptor is OK in author’s experimental condition, and second I wonder whether increased pERK by NAGly was reproduced if the detection method for pERK is same in McHugh’s paper like what reviewer 1 pointed out.

To the response for comment #4

Because the aim of this experiment is to enhance the surface expression of HA-GPR18, surface expression of pplss-3HA-hGPR18 should be compared with 3HA-hGPR18. If it is hard to establish the cells, alternatively you can speculate from present results how much the surface expression was changed by the addition of pplss, and include in result section.

To the response for other comments

I appreciate to the nice arrangements for others.

---

## Round 0.3 · accepted · Accept

Authors have revised and improved the manuscript.